# Effect of Salt Stress on Growth and Physiological Properties of Asparagus Seedlings

**DOI:** 10.3390/plants11212836

**Published:** 2022-10-25

**Authors:** Xin Guo, Naveed Ahmad, Shuzhen Zhao, Chuanzhi Zhao, Wen Zhong, Xingjun Wang, Guanghui Li

**Affiliations:** 1Institute of Crop Germplasm Resources (Institute of Biotechnology), Shandong Academy of Agricultural Sciences, Shandong Provincial Key Laboratory of Crop Genetic Improvement, Ecology and Physiology, Jinan 250100, China; 2Shandong Seed Administration Station, Jinan 250100, China

**Keywords:** asparagus, salt stress, growth, ion content, osmolyte, antioxidant enzyme activity

## Abstract

Salt stress could inhibit the growth and development of crops and negatively affect yield and quality. The objective of this study was to investigate the physiological responses of different asparagus cultivars to salt stress. Twenty days old seedlings ofasalt-tolerant Apollo andasalt-sensitive cultivar JL1 were subjected to 0 (CK) and120 mM NaCl stress for 20 d. Their changes in growth, ion contents, antioxidant enzyme activities and gene expression were analyzed. Salt stress significantly inhibited the growth of both cultivars, and JL1 showed a greater decrease than Apollo. The root development of Apollo was promoted by 120 mM NaCl treatment. The Na^+^ content in roots, stems, and leaves of both cultivars was increased under salt stress, while K^+^ content and K^+^/Na^+^ decreased. The salt-tolerant cultivar Apollo showed less extent of increase in Na^+^ and decrease in K^+^ content and kept a relatively high K^+^/Na^+^ ratio to compare with JL1. The contents of proline, soluble sugar and protein increased in Apollo, while thesesubstances changed differently in JL1 under salt stress. Activities of superoxide dismutase (SOD), peroxidase (POD), and catalase (CAT) were gradually increased under salt stress in Apollo, while the corresponding enzyme activities in JL1 were decreased at the late stage of salt stress. The expression of *SOD*, *POD*, and *CAT* genes of both cultivars changed in a similar way to the enzyme activities. Malondialdehyde (MDA) content was increased slightly in Apollo, while increased significantly in JL1. At the late stage of salt stress, Apollomaintained a relatively high K^+^/Na^+^, osmotic adjustment ability and antioxidant defense capability, and therefore exhibited higher tolerance to salt stress than that of JL1.

## 1. Introduction

Salinity is one of the major abiotic stresses worldwide for most plant species and negatively affects the yield and quality of crops [1]. More than 8% of the world’s land had been affected by salinity, and the area was predicted to continuously increase [2,3,4]. Breeding of salt-tolerant crops is an important option to efficiently utilize saline land [5]. Understanding the salttolerance mechanisms of different crops is important for the breeding of salt-tolerant crops.

Salt stress causesa series of physiological changes in plants, such as osmotic stress, ionic imbalance, gas exchange reduction and reactive oxygen species (ROS) accumulation [6]. Meanwhile, the plantevolved adaptation strategies to resist salt stress, such as osmotic regulation, ion homeostasis and the antioxidant defense system [7]. The accumulation of osmolytes, such as betaine, proline, soluble sugar, and soluble proteins under salt stress, plays an important role in maintaining the lower osmotic potential of plants and in relieving the pressure of osmotic stress [8]. Some studies showed thatproline plays a key role in osmotic adjustment. The content of proline is positively correlated with salt stress level, and salt-tolerant cultivars accumulate more proline [9,10]. However, other studies suggested that proline may contribute to ROS scavengingand membrane stabilization under salt stress [11,12].

Salt stress increases the uptake of Na^+^ and decreases the uptake of K^+^, and leading to disruption of ionic homeostasis, which inhibits the absorption of other nutrient ions, resulting in nutrient deficiency and ion toxicity [13,14]. K^+^/Na^+^ homeostasisplays an important role in maintaining the physiological activity of cells, and ion-homeostasis could be attained through various mechanisms such as selective ion absorption, Na^+^ efflux and ion regionalization [15]. There are two main pathways that regulate Na^+^ concentrations in the cytoplasm of plant cells. One is the tonoplast-localized Na^+^/H^+^ exchanger (NHX1), which is effective for Na^+^ detoxification by sequestration of Na^+^ within the vacuole; the other is plasma membrane-localized Salt Overly Sensitive (SOS) Na^+^/H^+^ antiporters, which attribute to export Na^+^ out of the cells [16]. Studies suggest that Na^+^ content and Na^+^/K^+^ of salt-tolerant cultivars are markedly lower than those of salt-sensitive cultivars under salt stress. Na^+^ content and Na^+^/K^+^ can be used as indicatorfor screening salt-tolerant cultivars [17,18].

Chloroplast, mitochondria and peroxisome produce excessive ROS under salt stress [19]. ROS can cause oxidative damage to lipids, act as toxic substances in the cell and as the signal of the stress response [20]. Plants eliminate ROS through two antioxidant defense systems, the enzymatic and non-enzymatic antioxidant systems [21]. The enzymatic antioxidant system mainly consists of superoxide dismutase (SOD),peroxidase (POD), catalase (CAT), ascorbate peroxidase (APX) and glutathione reductase (GR). Studies have shown that activities of the antioxidant enzyme such as SOD, POD and CAT increased in salt-tolerant cultivars but decreased in salt-sensitive cultivars under salt stress [22,23,24]. Antioxidant-related genes showed significant upregulation under salt stress, suggested that their roles in salt stress response and can be used as an indicator for breeding salt-tolerant cultivars [25]. MDA is the primary product of lipid peroxidation, which could inactivate the protein and enzyme, and disturb the normal structure and function of the cytomembrane. A lower concentration of MDA is usually observed in salt-tolerant cultivars than in salt-sensitive cultivars under salt stress [9,26].

Asparagus (*Asparagus officinalis* L.) belongs to the lily family, is a perennial and diecious vegetable popularly consumed worldwide. Asparagus contains a high level of nutrients and antioxidant compounds, and has positive effects on preventing hypertension, heart disease and certain cancers [27,28,29]. Asparagus originate in the Mediterranean region andis now cultivatedin many countries worldwide. Asparagus appears to be moderately resistant to several stress conditions. A low level of NaCl in soil showed a positive effect on spear yield [30,31]. Meanwhile, the spear yield and quality decreased when subjected to more than 0.3% NaCl of salt stress. Under salt stress, the differentially expressed genes are mainly enriched in ion regulatory pathways, antioxidant enzyme system, photosynthetic and carbon catabolism processes, revealing the key pathways in salt stress responses of asparagus [32]. However, little information is available on gene expression patterns, physiological and biochemical responses between asparagus cultivars with different salt tolerance under salt stress. The purpose of this study is to investigate the growth, ion uptake, osmolytes accumulation and antioxidative responses of salt-tolerant and salt-sensitive cultivars under salt stress at the seedling stage. Results from this study provide information to understand the physiological mechanism of salt tolerance, which could be a help for salt-tolerant asparagus breeding and cultivation. 

## 2. Materials and Methods

### 2.1. Plant Materials and Treatments

Apollo and JL1, two popular commercial cultivars in China, were used in this study. Apollo is a salt-tolerant cultivar. The seeds were supplied by Walker Bros Inc, USA. JL1 is a salt-sensitive cultivar. The seeds were obtained from Beijing Academy of Agriculture and Forestry Sciences, China. Our previous study showed that the germination rate of Apollo was 4.5 folds higher than JL1 under 120 mM NaCl stress (unpublished data).

The experiment was conducted in a greenhouse. Healthy asparagus seeds were first soaked for 48 h in sterilized distilled water at 30 °C and changing the water every 12 h. The soaked seeds were covered with wet gauze and germinated at 25 °C for 48 h in the incubator, and then sown in plastic pots (210 mm diameter, 160 mm height) filled with clean sand. Eight seeds were sown in each pot. The plants were grown in Hoagland solution for 20 days, then the pots were divided into two groups. One group continued the growth in the Hoagland solution, while 120 mM NaCl was added to the Hoagland solution in the other group of plants (stressed plants).For each treatment, 10 pots (six plants per pot), were used and watered daily with 500 mL Hoagland solution with or without NaCl. After 0, 12, 24, 36, and 48 h of salt stress, root materials were sampled and immediately put into liquid nitrogen and then stored at −80 °C for gene expression analyses. The roots used for antioxidant enzyme activity and osmolytescontent analysis were collected at 2, 4, 6, 8 d after salt stress. K^+^ and Na^+^ contents were determined at 7 and 15 d after salt treatment. Plants used for biomass and phenotypic analysis were harvested at 20 d after salt stress.

### 2.2. Measurement of Plant Height and Biomass

Plant height was measured from the soil surface to the top of the plant. The roots and shoots of the seedlings were collected separately and placed in envelops. The dry weight of the roots and shoots were determined after dryingat 105 °C for 30 min then at 80 °C for 48 h. The relative value was calculated for plant height, root dry weight and biomass per plant by comparing values under salt stress treatment to values of CK, using the following formula:(1)$$\mathrm{The}\;\mathrm{relative}\;\mathrm{value}=\frac{\mathrm{Value}\;\mathrm{of}\;\mathrm{salt}\;\mathrm{stress}\;\mathrm{treatment}}{\mathrm{Value}\;\mathrm{of}\;\mathrm{non}\;\mathrm{stress}\;\mathrm{treatment}}\times 100\%$$

### 2.3. Determination of Na^+^, K^+^ and K^+^/ Na^+^

The dry samples were ground into powder and pass through a 40-mesh screen. Then, 0.5 g powder was placed in a crucible to ash in a muffle furnace. The ash was dissolved in 5 mL of 0.5 mol·L^−1^ HNO_3_, then transferred to a 100 mL volumetric flask and topped up with deionized water. Na^+^ and K^+^ concentrations were measured by atomic absorption spectrophotometer (PYESP9-400, Waltham, MA, USA).

### 2.4. Antioxidant Enzymesactivity and MDA Content

Rootsamples (0.5 g) were homogenizedusing a precooled mortar in 50 mM potassiumphosphate buffer (pH 7.8) containing 1% (*w*/*v*) PVP at 0~4 °C. The homogenate was filtered through two layers of filter paper and centrifuged at 10,000× *g* for 20 min at 4 °C. The supernatant was used for enzyme activity analysis.

SOD activity was measured by the method of Giannopolitis and Ries [33]. The reaction mixture consisted of 50 mM phosphate buffer (pH 7.8), 13 mM methionine, 75 mM NBT, 0.1 mM EDTA and 2 mM riboflavin. Reactions with 50 µL enzyme extract and 3 mL reaction mixture were carried out in a light incubator under a light intensity of 4000 Lux for 30 min. Oneunit of SOD was defined as the amount of enzyme which causes 50% inhibition of the NBT reduction. The reduction of NBT was measured by an ultraviolet spectrophotometer at 560 nm.

POD activity was determined based on guaiacol colorimetric method. Reaction mixture contained 50 mL 100 mM potassium phosphate (pH 6.0), 30 µL 0.3 mM guaiacol, and 20 µL 30% H_2_O_2_. 20 μL enzyme solution and 3 mL reaction mixture were added into the colorimetric cup to start the reaction. Colorimetry at 470 nm was performed at 30 s intervals for a total of 5 readings. The activity of the POD enzyme was expressed by the change value of absorbance per minute [34].

CAT activity was measured according to Aebi [35]. 50 μL enzyme solution was added into 3 mL reaction system (2.4 mL 100 mM potassium phosphate (pH 7.0), 0.6 mL 100 mM H_2_O_2_). Colorimetry at 240 nm was performed at 30 s intervals for a total of 5 readings. The activity of the CAT enzyme was expressed by the reduction of absorbance per minute.

MDA was assayed by the thiobarbituric acid reaction method [36]. Frozen sample of 0.5 g was homogenized in 0.1% (*w*/*v*) trichloroacetic acid (TCA) solution. The homogenate was centrifuged at 12,000× *g* for 10 min. 1 mL supernatant was added to 20% TCA (2 mL) containing 0.6% thiobarbituric acid (TBA) in a clean glass tube. The mixture was heated in a water bath at 90 °C for 30 min, cooled on ice immediately, and then centrifuged at 4000× *g* for 10 min. The absorbance was recorded at 600, 532 and 450 nm. 

### 2.5. Determination of Proline, Soluble Sugar and Soluble Protein

Proline content was determined by the Bates method [37]. Fresh samples (0.2 g) were homogenized in a pre-chilled mortar using 5 mL (3%) sulfosalicylic acid and centrifuged at 15,000× *g* for 15 min. 0.5 mL of the supernatant was transferred to a test tube and mixed with 1 mL sulfosalicylic acid (3%), 1 mL glacial acetic acid (99.5%), 2 mL acid ninhydrin (2.5%). The tubes were incubated in a boiling water bath for 1 h, then transferred to an ice bath to terminate the reaction. 4 mL of toluene (99.5%) was added to the tubes and mixed thoroughly for 30 s. After standing, the upper layer toluene was collected to measure the absorbance at 520 nm in a UV-spectrophotometer. Proline content was determined using the proline standard curve.

Soluble sugar was determined by the anthrone colorimetry method [38]. Then, 0.5 g of chopped fresh samples were placed in a test tube and then 10 mL distilled water was added. The tubes were heated in boiling water for 1 h, and then centrifuged at 5000× *g* for 10 min. The reactionmixture containing 1 mL supernatant and 5 mL anthrone (100 mg anthrone + 100 mL 80% H_2_SO_4_) was heated at 100°C for 10 min, and then absorbance was read at 620 nm.

Soluble protein was determined by Coomassie brilliant blue method [39]. 0.5 g samples were ground into a homogenate using 5 mL of 50 mM phosphate buffer (pH 7.8). The homogenate was centrifuged at 12,000× *g* for 20 min. 50 µL of the supernatant was transferred to a test tube and 4 mL Coomassie brilliant blue (0.01%) and 1 mL distilled water were added. After standing for 3 min, absorbance was read at 595 nm in a UV-spectrophotometer, distilled water with Coomassie brilliantblue was used as the blank control. Protein content was determined using the BSA standard curve.

### 2.6. RNA Isolation and Quantitativert-PCR

Total RNA extraction was carried out using a Trizol Reagent kit (TaKaRa, Inc., Dalian, China) according to the manufacturer’s instructions. RNA samples were treated with DNase I to remove genomic DNA contamination. RNA quality and purity were analyzed by Agilent 2100 and NanoDrop. Total RNA from each sample was used to enrich mRNA with Oligo (Dt) and then cleaved into short fragments (~200 nt) in a fragmentation buffer. Through reverse transcription using random hexamer primer to get the first-strand cDNA and synthesis of the second strand of cDNA using buffer, dNTPs, RNaseH, and DNA polymerase. After end repairing and adding sequencing adaptors, the cDNA fragments were subjected to PCR amplification using gene-specific primers. Ubiquitin-40S ribosomal protein S27a was used as an internal reference gene. The primers used for qRT-PCR are as follows (Table 1). The qRT-PCR was performed using SYBR Green Pro Taq HS premixed qPCRkit in a 7500 Real Time PCR System. The parameters of the thermal cycle were 94 °C for 10 min, followed by 40 cycles of 94 °C for 15 s and 60 °C for 1 min in a 20 mL volume. Three biological replications were performed for each reaction with actin gene as an internal reference. The relative gene expression level was calculated by 2^−ΔΔCT^ method.

### 2.7. Statistical Analysis

Analysis of variance (ANOVA) was conducted with SPSS software 21.0 (IBM Corp, Armonk, NY, USA) based on three replicates. Means were compared using the least significant difference (LSD) test at a *p* < 0.05 threshold. Figures were plotted using Sigma Plot 10.0 software (Systat Software Inc., Chicago, IL, USA).

## 3. Results

### 3.1. Seedling Growth

Salt stress treatment adversely affected the plant growth of both cultivars, and this negative effect was moresevere in JL1 than in Apollo (Table 2 and Figure 1). At 20 days after salt stress, the relative plant height was 80% in Apollo, and 55% in JL1 under salt stress as compared to the control. The root growth showed a slight decrease in JL1, while a slight increase in Apollo was observed under salt stress. The relative biomass was 96% in Apollo, and 71% in JL1 under salt stress to compare with the control, reflecting the greater salt tolerance of Apollo compared to JL1. The foliage etiolation and defoliation of cladophylls were observed in JL1 under salt stress, however, these effects were not observed in Apollo (Figure 1).

### 3.2. Content of K^+^ and Na^+^

The K^+^ and Na^+^ contents were determined at 7 and 15 days of treatments. Both cultivars exhibited a decline in K^+^ content in the root, shoot and cladophyll during salt stress (Figure 2). To compare with control, K^+^ content in the root, shoot and cladophyll decreased 7.83%, 27.56%, 16.59% in Apollo, and decreased 4.72%, 14.53%, 13.78% in JL1 after salt stress for 7 d, respectively. The K^+^ content was decreased with prolonged salt stress. K^+^ content of root, shoot and cladophyll decreased 16.61%, 28.41%, 24.79% in Apollo, and decreased 22.69%, 32.13%, 26.50% in JL1 as compared to control, respectively. 

Plant Na^+^ content of two cultivarsincreased significantly under salt stress as compared to control (Figure 2). The content of Na^+^ was found to continuously increase with the duration of salt stress in JL1, however, Apollo showed a stable increase proportion. After 15 d of salt stress, Na^+^ content of root, shoot and cladophyll increased 60.47%, 74.64%, 99.89% in Apollo, while increased 121.24%, 158.62%, 97.13% in JL1 as compared to control, respectively.

K^+^/Na^+^ showed a significant decrease in the two cultivars during salt stress, and a greater reduction in JL1 than Apollo was observed. With salt stress time extension, a continuous decrease in K^+^/Na^+^ was seenin JL1, however, K^+^/Na^+^ was almost stable in Apollo.

### 3.3. Antioxidant Enzymes Assay

The salt stress-induced changes of antioxidant enzymes activity were followed after 2, 4, 6 and 8 days of treatment. In comparison with control, the antioxidant enzymatic activity of SOD, POD and CAT showed a significant increase under salt stress, which increased gradually in Apollo, whereas decreased with extending time of salt stress were detected in JL1 (Figure 3). The SOD, POD and CAT activities of Apollo gradually increased from 2 d to 8 d of salt stress. The maximum increases were observed on 8 dafter salt stress, increasing 80.57%, 86.22% and 141.21%, respectively (Figure 3a–c). The maximum increase (87.01%) of SOD in JL 1was observed on 6 d after salt stress as compared to control (Figure 3d). The maximum increase (48.33%) of POD in JL1 was seen on salt stress for 4 d. However, that decreased by 11.84% compared to control after 8 d of salt stress (Figure 3e). After 2 d of salt stress, the CAT activity in JL1 was 39.91% higher than that of the control, while after salt treatment for 8 d, CAT activity was only 4.83% higher than that of the control (Figure 3f).

### 3.4. MDA Content

The MDA contents were continuously increased in two cultivars during salt stress as compared to the control, and JL1 showed a higher increase than Apollo (Figure 4a). The MDA content of Apollo increased 18.62% to 26.59% during salt stress from 2 d to 8 d. However, the MDA content of JL1 showed a much more increase of 42.48% to 125.49% during salt stress from 2 d to 8 d as compared to the control, respectively (Figure 4b).

### 3.5. The Content of Osmolytes 

The content of proline, soluble sugar and soluble protein in the two cultivars showed a significant increase under salt stress as compared to the control. However, with the duration of treatment, the changing patterns of two cultivars were different (Figure 5). The changing patterns of proline contents were similar in two cultivars, all increased remarkably with the duration of salt stress. The proline contents of Apollo and JL1 increased 9-fold and 26-fold after 8 d exposure to salt stress. In Apollo, the soluble sugar content increased continuously from 2 d to 8 d of salt treatment. The soluble sugar content was 150.52% higher than that of the control after 8 d exposure to salt stress. In JL1, a maximum increase (114.79%) of soluble sugar content was observed after salt stress for 6 d, and then decreased. The changing patterns of the soluble protein of the two cultivars were similar to that of the soluble sugar. Soluble protein content of Apollo increased 33.95%% after 8 d exposure to salt stress. However, there was little difference in JL1 between control and salt stress. 

### 3.6. Gene Expression Analysis Using qRT-PCR

The expression of *SOD* increased from 12 h of salt treatment in both cultivars, and the peak level was observed at 24 h of treatment. At the time, the expression levels of *SOD* in Apollo and JL1 were upregulated 1.66-fold and 1.99-fold to compare with the control, respectively. At 36 h of treatment, the expression of *SOD* started to decrease. At 48 h of treatment, the expression of *SOD* in JL1 was only 13.27% of the control, while the expression level in Apollo was 72.81% of the control. The maximum upregulation of *POD* was observed at 12 h after salt stress, the expression levels in Apollo and JL1 upregulated 3.99-fold and 3.38-fold to compare with control, respectively. At 48 h of treatment, the expression of *POD* in JL1 was only 17.18% of the control, while the expression level in Apollo was 1.60-fold of the control. Compared to control, the upregulation of *CAT* gene in Apollo increased gradually during 48 h salt stress, whereas JL1 showed maximum upregulation at 36 h and significant down-regulation at 48 h exposure to salt stress (Figure 6).

## 4. Discussion

In the present study, salt stress decreased the growth of both cultivars, but the adverse effects of salt were much more severe in JL1 than in Apollo in terms of biomass, physiological response and gene expression. Our results suggested that Apollo is a relatively salt-tolerant cultivar compared to JL1. The reduction in growth of bothcultivars under salt stress in our study confirmed previous conclusions that asparagus was a relatively salt-tolerant crop [30]. In this study, more inhibition was observed in shoots than that in roots in both cultivars by salt stress. Similar results were found in rice, more growth reduction was found in leaves than that in roots under salt stress [40]. 

The increased Na^+^ accumulation in plant cells induce ion toxicity. K^+^ plays an important role in maintaining ion homeostasis and regulating plant growth and development under salt stress [13]. In the current study, Na^+^ accumulation was found increased significantly, while K^+^ and K^+^/Na^+^ declined after salt stress. Similar results were found in rice and potato. These results suggested that Na^+^ and K^+^ accumulation in a salt-tolerant cultivar is similar to that in a salt-sensitive cultivar under salt stress [41,42]. In our study, the K^+^/Na^+^ showed a continuous decline in JL1 with increased salt stress time, whereas Apollo exhibited a less change, a higher K^+^/Na^+^ was maintained in Apollo than JL1. This might suggest that the salt-tolerant cultivar could quickly modulate the uptake of Na^+^ and K^+^ to adapt to salt stress, while the salt-sensitive cultivar could not.Our results are inconsistent with peanut and Chinese cabbage [18,26]. These results suggestedthat there is a positive relationship between K^+^/Na^+^ and salt tolerance. K^+^/Na^+^ can be used as an important reference to screen salt tolerance cultivars [43].

High concentrations of salt result in oxidative stress and induce a series of cell damage, meanwhile, the antioxidant protection system of plants could minimize the damage [20]. The antioxidative enzymes SOD, POD and CAT could cooperatively and efficiently scavenge the active oxygen species (ROS)and keep the ROS below toxic level [44]. In the present research, salt stress activated the antioxidative enzymes of SOD, POD and CAT of in both cultivars. The activation was more obvious in the salt-tolerant cultivar than the salt-sensitive one. Similar results have been observed in wheat [45] and rice [46]. SOD, POD and CAT activities of salt-tolerant cultivars were enhanced more under salt stress than that of sensitive cultivars. However, pea leaves SOD activity showed a significant decline under salt stress [47]. It may be responsible for the different sensitivity between species to salt stress or oxidative stress [42]. In our study, three antioxidant enzyme activities in Apollo gradually increased from 2 d to 8 d after salt stress, however the enzyme activities in JL1 showed a decreased trend at late salt stress compared with the early salt stress period. It may be attributed to that the sustained stress exceed the ROS scavenging capacity of salt-sensitive cultivar [48]. We further detected the expressionlevel of antioxidant enzyme genes including *SOD*, *POD* and *CAT*. The expressions of *SOD*, *POD* and *CAT* were significantly upregulated under salt stress. The upregulation in Apollo was higher than that in JL1, which were almost consistent with their enzyme activities. Similar results were found in *Lotus japonicas* and tobacco [49,50]. Transcriptomic analyses of asparagus revealed that antioxidant system components were induced under salt stress. Genes encodingantioxidase were upregulated at different stages of salt stress, suggesting that the enzymatic pathway may play important roles in protecting asparagus against oxidative damage under salt stress [32]. Our study confirmed that the antioxidant enzyme gene expression level was closely related to the salt tolerance of asparagus cultivars.

MDA, an indicator for evaluation of lipid peroxidation or degree of cellmembrane damage, was increased under stress condition. In the present research, MDA content was significantly increased during salt stress, and JL1 accumulated more MDA than Apollo. Our results indicated that a more effective enzymatic antioxidant system in Apollo conferred higher ROS scavenging capacity to control MDA production. Similar results were found in eggplant [51]. Our study also confirmed that activating antioxidant enzymes under salt stress cannot completely eliminate ROS caused cell damage as suggested by previous studies [26,48]. Higher activity of antioxidaseand less accumulation of MDA under salt stress explain the better tolerant of Apollo compared to that of JL1.

The accumulation of organic compounds under salt stress plays animportant role in osmotic adjustment in plants. Proline is a kind of hydrophilic macromolecules. There are disputes about the role of proline in osmotic regulation. It has been proposed that the accumulation of proline played a key role in osmotic adjustment and enhanced salt tolerance of maize inbred lines [9]. A contrasting view reported that proline was not directly involved in alleviating osmotic stress, but acted as a ROS scavengerand membrane structures stabilizer to resist stress condition [52]. In the present study, proline accumulation showed a higher increase in JL1 than that in Apollo during salt stress, suggesting that the proline was not positively correlated to salt tolerance in asparagus. Similar results in rice also suggested that a higher level of proline was observed in salt-sensitive cultivars than salt-tolerant cultivars under salt stress. In this case, the rice root growth was severely inhibited insalt-sensitive cultivars, but no significant inhibition was observed in salt-tolerant cultivars [53,54]. The soluble sugar and soluble protein play a key role toosmotic adjustment and preserving membrane integrity under stress conditions. Under salt stress, the total soluble sugar increased significantly in rice and was closely related to the osmotic adjustment ability of cultivars [55]. In the present study, Apolloaccummulated more soluble sugar and soluble protein compared to JL1 under salt stress. Similar with our findings, higher soluble sugar and soluble protein accumulations were observed insalt-tolerant chickpeas [56] and tomatoes [57] than in the salt-sensitive onesunder salt stress. The soluble sugar and soluble protein gradually increased in Apollountil 8 d after salt stress, but the maximum accumulation in JL1 on 6 d after salt stress. The results suggested that the greater osmotic adjustment capacity of Apollo may explain its higher salt-tolerant compared with JL1.

## 5. Conclusions

Our results demonstrated that the growth of salt-sensitive cultivar JL1 showed more inhibition than the salt-tolerant cultivar Apollo under salt stress. Apollo could keep higher K^+^/Na^+^ as compared to JL1 at the late stage of salt stress. The osmolytes content and antioxidant enzyme activities were gradually increased under salt stress in Apollo, which were decreased in JL1 at the late stage of salt stress. The ability to keep ion balance, accumulate more osmolytes and the upregulated ROS scavenging enzyme activities are important mechanisms that may explain the salt tolerance of Apollo. 

## Figures and Tables

**Figure 1 plants-11-02836-f001:**
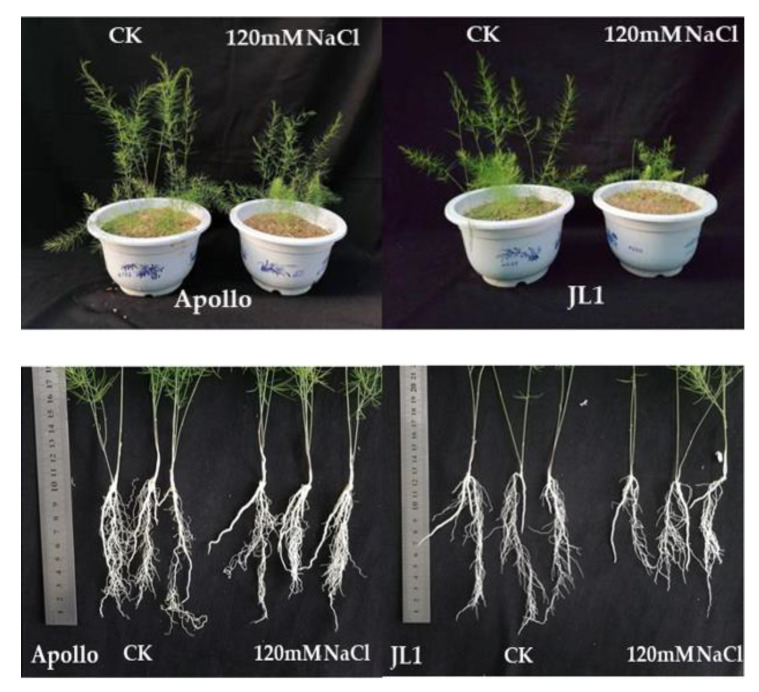
The phenotype of two asparagus cultivars under CK and salt stress.

**Figure 2 plants-11-02836-f002:**
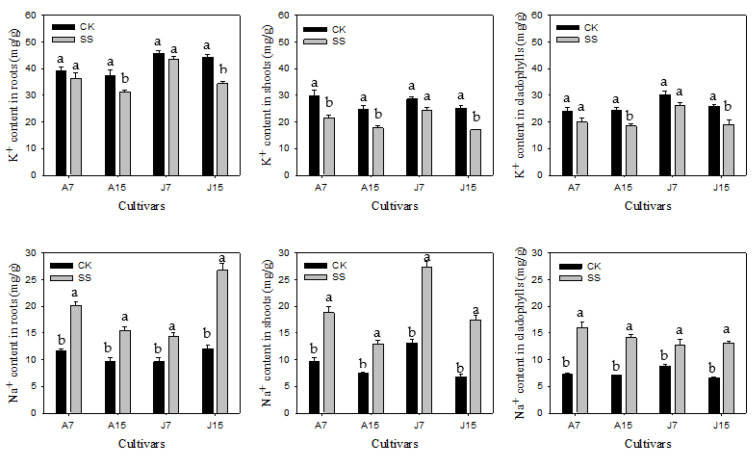
K^+^, Na^+^ and K^+^/Na^+^ of two asparagus cultivars under 7 d and 15 d after salt stress. A and J represent Apollo and JL1,7 and 15 mean the days after salt stress, respectively. Different lowercase letters in the bar graph indicate significantly different from the control at *p* < 0.05. CK, meaning seedlings were watered by Hoagland solution without NaCl, and SS, meaning seedlings were subjected to salt stress of 120 mM NaCl.

**Figure 3 plants-11-02836-f003:**
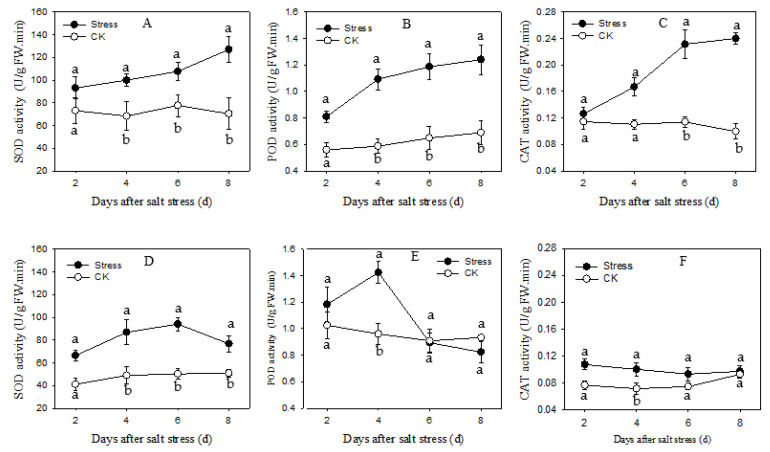
The SOD, POD, CAT enzyme activities of two asparagus cultivars under salt stress. (**A**–**C**) are Apollo and (**D**–**F**) are JL 1. Different lowercase lettersin the line graph indicate significantly different from the control at *p* < 0.05. CK meaning seedlings were watered by Hoagland solution without NaCl, and SS meaning seedlings were subjected to salt stress of 120 mM NaCl.

**Figure 4 plants-11-02836-f004:**
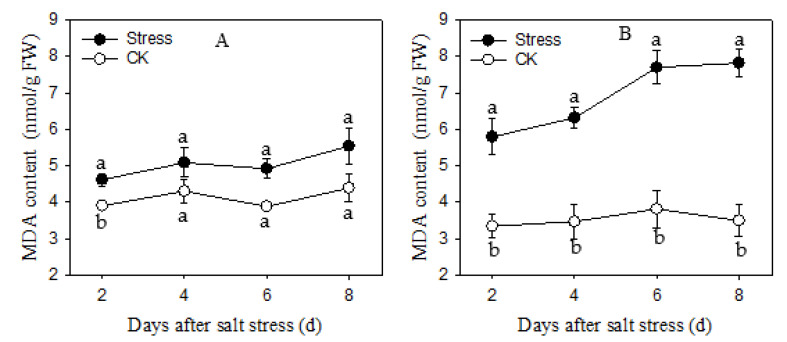
MDA content of two asparagus cultivars under salt stress. (**A**) is Apollo and (**B**) is JL 1.Different lowercase letters in the line graph indicate significantly different from the control at *p* < 0.05. CK meaning seedlings were watered by Hoagland solution without NaCl, and SS meaning seedlings were subjected to salt stress of 120 mM NaCl.

**Figure 5 plants-11-02836-f005:**
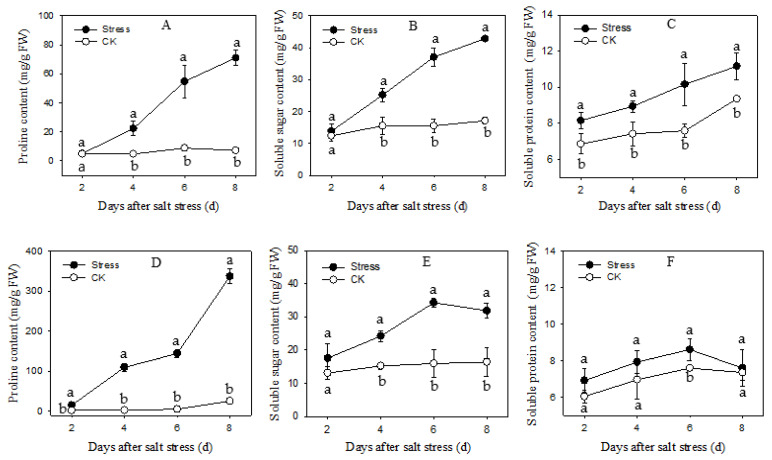
The osmotic regulation substances content of two asparagus cultivars under salt stress. (**A**–**C**) are Apollo and (**D**–**F**) are JL 1. Different lowercase letters in the line graph indicate significantly different from the control at *p* < 0.05. CK meaning seedlings were watered by Hoagland solution without NaCl, and SS meaning seedlings were subjected to salt stress of 120 mM NaCl.

**Figure 6 plants-11-02836-f006:**
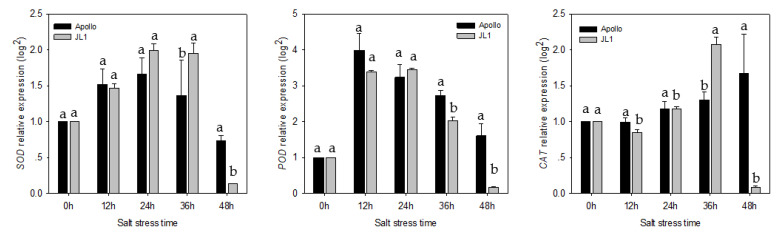
The *SOD*, *POD*, *CAT* gene relative expression in two cultivars under salt stress. Different lowercase letters in the bar graph indicate significantly different of gene relative expression between the two cultivars at *p* < 0.05.

**Table 1 plants-11-02836-t001:** The following primers were used in this experiment.

Gene Name	Gene ID	Primer Name	Primer Sequence(5’-3’)
ubiquitin-40Sribosomal protein S27a	LOC109820108	LOC109820108-F	CAATGTCAAGGCCAAGATCC
		LOC109820108-R	CTTCTGGATGTTGTAGTCGG
Superoxidedismutase [Cu-Zn]	LOC109836512	LOC109836512-F	CATCATCAGACCTTGAGCAG
		LOC109836512-R	AGGAGGAGAAATTAGGGTTAGG
peroxidase 12-like	LOC109839605	LOC109839605-F	CTCTCCTCTCATCATCTACAC
		LOC109839605-R	CTCTCCTCTCATCATCTACAC
catalase isozyme 1-like	LOC109837240	LOC109837240-F	TCACTCACGATGTTTCTCAC
		LOC109837240-R	TCAATGTTTCAGGACTCCCA

**Table 2 plants-11-02836-t002:** Effects of NaCl stress on plant height and biomass of two asparagus cultivars.

Cultivars	Treatment	Plant Height(cm)	Root Dry Weight Per Plant (g)	Biomass Per Plant (g)
Apollo	CK	31.84 ± 2.54 a	0.17 ± 0.01 a	0.32 ± 0.01 a
	Stress	25.40 ± 2.52 b	0.18 ± 0.02 a	0.31 ±0.02 a
	Relative value	80%	106%	96%
JL1	CK	30.82 ± 2.33 a	0.08 ± 0.01 a	0.20 ± 0.03 a
	Stress	17.02 ± 0.86 b	0.07 ± 0.02 a	0.14 ± 0.02 b
	Relative value	55%	79%	71%

Data are means ± SE of three replicates each containing five grown plants independently. Lower case letters indicate differences (*p* ≤ 0.05) between treatments.

## Data Availability

Data are contained within the article.

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
