# Peer review of "Effect of Salt Stress on Growth and Physiological Properties of Asparagus Seedlings"

_plants, 2022, doi:10.3390/plants11212836_

Round 1
Reviewer 1 Report
Thank you for getting the opportunity to review: “ Effect of salt stress on growth and physiological properties of 2 asparagus seedlings” by Xin Guo, Naveed Ahmad, Shunzhen Zhao, Chuanzhi Zhao and Wen Zhong, Xingjun Wang and Guanghui Li.
The paper provides a description and insight in how a salt sensitive and a more salt tolerant asparagus cultivar react physiological on salt stress on young seedlings.
Following is some points for reflection and for updating the paper accordingly:
· when using young plants – is there any data or papers which look into if the salt stress reaction be the same if performed with older (grown for several years) plants. Or is the data solely related to the establishment of the plants
· line 304 - wasn't it known that JL1 was salt sensitive from the start of this research. It is indicated so in the start of this paper.
· line 305 - how can it be deducted that the asparagus are salt tolerant - when the growth is reduced. - Is this due to that other plants, cultivars or species, would be more severely affected?
· For the conclusion - any reflection of what impact these results will have on acquiring more knowledge to salt stress. As stated to be one of the objectives of this paper - stated in line 90-92. One way would be to highlight the suggestions from the discussion.
· line 376 - As a salt sensitive cultivar was used for the comparison - it isn't a surprise that the growth are more inhibited. However, here it is the conclusion of the paper
There are some general comments regarding the writing and description of the legends.
Please go through the text again to update the text, here are some examples of places in the text which need to be rephrased or word added to help the reader to get the full meaning of the sentences.
· please revise the sentence line 55 to 58
· in line 124 - "by precooled" could be exchanged by - "using a precooled"
· It could be a help to the reader to add the time point when this is measured - line 195 to 203
· in line 197, "respectively" can be omitted
· for line 238- 239, please revise this part of the sentence, so it is more clear which trend it refers to
· line 319 - guess a word is missing her - "the uptake of Na+ and K+ adapt to salt stress" and instead - "the uptake of Na+ and K+ TO adapt to salt stress"
· Normally Cdna are written "cDNA". Materials and methods 2.6
· The paragraph title of results 3.2 - is the changes due to uptake? or should the title be “content of K+ and NA+”
· In general, it should be possible to read the legends of tables and figures without reading the whole text. Please go through these and e.g. add the definition of CK, stress, and where relevant add the time point after stress for these data.
· The paper operates with both NA+/K+ and K+/NA+ - if this is not two sides of the same - please provide a description of these two ways to view the ratio. In the material and methods - in 2.3 the focus is on the K+/NA+

Author Response
Following is some points for reflection and for updating the paper accordingly:
when using young plants – is there any data or papers which look into if the salt stress reaction be the same if performed with older (grown for several years) plants. Or is the data solely related to the establishment of the plants
Reply: In most of the previous studies, the asparagus seedlings were used to research salt-tolerant mechanisms. Researches using older asparagus were rarely reported. Salt tolerance differences between seedlings and older plants need to further study.
line 304 - wasn't it known that JL1 was salt sensitive from the start of this research. It is indicated so in the start of this paper.
Reply: We previously screened salt tolerance of dozens of asparagus cultivars. The relative growth under salt stress of JL1 was lower than other cultivars, it was considered as a salt sensitive cultivar or cultivars with low tolerance to salt.
line 305 - how can it be deducted that the asparagus are salt tolerant - when the growth is reduced. - Is this due to that other plants, cultivars or species, would be more severely affected?
Reply: The growths of all asparagus cultivars were restrained under salt stress. The suppressions were slight under moderate salt stress. Under 0.7% NaCl, the suppressions of different cultivars had significant differences, the growth of salt-tolerant cultivars inhibited less, whereas salt sensitive cultivars inhibited more. Previous research showed that shown that germination rate of asparagus seeds was not suppressed by soil salinity of 0.5%, disease index of asparagus roots was reduced after addition of NaCl to asparagus field and the yield increased. Field production showed that the spear yield was not decreased in the asparagus field of 0.3% salinity content, however the yields of most crops e. g. wheat, maize and peanut significantly decreased at the same salinity stress.
For the conclusion - any reflection of what impact these results will have on acquiring more knowledge to salt stress. As stated to be one of the objectives of this paper - stated in line 90-92. One way would be to highlight the suggestions from the discussion.
line 376 - As a salt sensitive cultivar was used for the comparison - it isn't a surprise that the growth are more inhibited. However, here it is the conclusion of the paper
Reply:More growth inhibition for the sensitive cultivars is the direct result of the study. The physiological response of Apollo was different from the sensitive cultivars. The ability to keep ion balance, accumulate more osmolytes and the up-regulated ROS scavenging enzyme activities are important mechanisms that may explain the salt tolerance of Apollo. This is the conclusion of our study.
There are some general comments regarding the writing and description of the legends.
Please go through the text again to update the text, here are some examples of places in the text which need to be rephrased or word added to help the reader to get the full meaning of the sentences.
Reply:We have revised and improved the language of the manuscript.
please revise the sentence line 55 to 58
Reply:We have revised the sentence.
in line 124 - "by precooled" could be exchanged by - "using a precooled"
Reply: Thank you for the suggestion! We corrected it in the manuscript.
It could be a help to the reader to add the time point when this is measured - line 195 to 203
Reply: We have added the details.
in line 197, "respectively" can be omitted
Reply: This is done in the manuscript.
for line 238- 239, please revise this part of the sentence, so it is more clear which trend it refers to
Reply: We have revised the sentence.
line 319 - guess a word is missing her - "the uptake of Na+ and K+ adapt to salt stress" and instead - "the uptake of Na+ and K+ TO adapt to salt stress"
Reply: We have revised the sentence.
Normally Cdna are written "cDNA". Materials and methods 2.6
Reply: We have revised it in the revised manuscript.
The paragraph title of results 3.2 - is the changes due to uptake? or should the title be “content of K+ and NA+”
Reply: Thank you for the suggestion! We have revised in the manuscript.
In general, it should be possible to read the legends of tables and figures without reading the whole text. Please go through these and e.g. add the definition of CK, stress, and where relevant add the time point after stress for these data.
Reply: According to the suggestions, we have added the definition about legends in the revised manuscript.
The paper operates with both NA+/K+ and K+/NA+ - if this is not two sides of the same - please provide a description of these two ways to view the ratio. In the material and methods - in 2.3 the focus is on the K+/NA+
Reply: Maintaining a high K+/Na+ in plants can alleviate the growth inhibition phenomenon caused by K+ deficiency when cells suffer from salt damage. Therefore, this paper chooses K+/Na+ to express the effect of salt stress on it. We removed all expression of Na+/K+.

Reviewer 2 Report
Material and methods
What are the conditions of the greenhouse?
For the random hexamer primer, explain more
Results
What is the formula used to calculate the relative value %?
L259 not in the proper position
The asterisks in figure2 are not in its correct position. Check the figure
Discussion
Explore more the result of Gene expression analysis
Conclusion
the conclusion is poorly written, it should be improved
Author Response
Material and methods
What are the conditions of the greenhouse?
Reply: It is glass greenhouse. The temperature adjusting by air vents above the greenhouse and air conditioner was 25~32 ℃ during the day and 15~20 ℃ at night.
For the random hexamer primer, explain more.
: Ubiquitin-40S ribosomal protein S27a was used as an internal reference gene. The primers used for qRT-PCR are as follows (Table 1). This primer sequences were added in the revised version.
Table 1: The following primers were used in this experiment
Gene ID |
primer name |
primer sequence(5'-3') |
LOC109820108 |
LOC109820108-F |
CAATGTCAAGGCCAAGATCC |
|
LOC109820108-R |
CTTCTGGATGTTGTAGTCGG |
LOC109836512 |
LOC109836512-F |
CATCATCAGACCTTGAGCAG |
|
LOC109836512-R |
AGGAGGAGAAATTAGGGTTAGG |
LOC109839605 |
LOC109839605-F |
CTCTCCTCTCATCATCTACAC |
|
LOC109839605-R |
CTCTCCTCTCATCATCTACAC |
LOC109837240 |
LOC109837240-F |
TCACTCACGATGTTTCTCAC |
|
LOC109837240-R |
TCAATGTTTCAGGACTCCCA |
- Results
What is the formula used to calculate the relative value %?
Reply: The relative value was calculated for plant height, root dry weight and biomass per plant by comparing values under salt stress treatment to values of CK, using the following formula: Please see the attachment.
L259 not in the proper position
Reply: We have adjusted the position.
The asterisks in figure2 are not in its correct position. Check the figure
Reply: The asterisks errors in figure2 are due to text composition, which have been corrected in the manuscript. Asterisks errors in figures have also been corrected.
- Discussion
Explore more the result of Gene expression analysis
Reply: We added more discussion in the revised version.
Transcriptomic analyses of asparagus revealed that antioxidant system components were induced under salt stress. Genes encoding antioxidase were up-regulated at different stages of salt stress, suggesting that the enzymatic pathway may play important roles in protecting asparagus against oxidative damage under salt stress [32]. Our study confirmed that antioxidant enzyme gene expression level was closely related to salt tolerance of asparagus cultivars.
- Conclusion
the conclusion is poorly written, it should be improved
Reply: a revised conclusion was provided as the following.
Our results demonstrated that the growth of salt-sensitive cultivar JL1 showed more inhibition than the salt-tolerant cultivar Apollo under salt stress. Apollo could keep higher K+/Na+ as compared to JL1 at late stage of salt stress. The osmolytes content and antioxidant enzyme activities were gradually increased under salt stress in Apollo, which were decreased in JL1 at late stage of salt stress. The ability to keep ion balance, accumulate more osmolytes and the up-regulated ROS scavenging enzyme activities are important mechanisms that may explain the salt tolerance of Apollo.

Reviewer 3 Report
In this work, the authors evaluated the effects of salinity on the physiological and molecular performance of two asparagus cultivars, one salt-resistant (Apollo) and one salt-sensitive (JL1). The authors used different approaches including measuring molecular markers and monitoring the performance of the antioxidant system at transcripts and enzymatic levels. The work is sounded and timely. However, I have some questions and concerns that need further explanation before recommending it for publication. On an additional note, the discussion section was nicely written.
Abstract
· It is more reliable to use molarities instead of percentages. The authors should consider changing it
· What are the main conclusions of this research?
Introduction
· L37. Please revise the grammar for a better understanding
· L42. Do only plants evolve an antioxidant defense system? Please clarify it
· L43. What is this role?
· L51. What is this important role?
· L54. “Two mainly pathways that regulate Na+ concentrations in cytoplasm of plant cells” is a sentence fragment. Please rewrite
· L73. Which proteins could MDA inactivate?
Methodology
· L104. Why did the authors choose 0.7 % of NaCl?
· L179, L180, L181 and elsewhere. Please write cDNA correctly
· L187. What is the sequence of the primers used in this study? What was the PCR efficiency?
Results
· The “X” and “Y” legends for the figures are somehow small
· L233-234. The sentence needs to be relocated
· The figure legends are consistently too short. Additional details and explanations are needed.
· Figure 6. I could not identify the control group.
Discussion
· The results of this study weres very nicely discussed
· L352. What is this role?
Conclusions
· What are the future directions or additional needed research in the area?
Author Response
1.Abstract
It is more reliable to use molarities instead of percentages. The authors should consider changing it
Reply: Thank you very much for your suggestion! However, the percentages were used to screen appropriate salt stress concentrations and evaluate cultivars for salt tolerance in our previous research. To be consistent, this percentage concentration of salt stress was still used in this paper.
What are the main conclusions of this research?
Reply: The physiological response of salt-tolerant cultivar Apollo might alleviate ion toxicity, osmotic stress, oxidative damage, and allow plants to be able to keep metabolic balance under constant salt stress. From the comparison of the salt sensitive and salt tolerant cultivars, our results demonstrated that keeping ion balance, accumulation of osmolytes and the antioxidant enzyme activity are key factors for asparagus to tolerant salt stress.
2.Introduction
L37. Please revise the grammar for a better understanding
Reply: Thank you very much for your suggestion! We checked the grammar and improved the language in the revised version of this manuscript.
L42. Do only plants evolve an antioxidant defense system? Please clarify it
Reply:The antioxidant defense system is an important regulatory mechanism for animals and plants to adapt to adverse environment
L43. What is this role?
Reply:The accumulation of osmolytes, such as betaine, proline, soluble sugar, and soluble proteins of plant cell can maintain the lower osmotic potential of plants, which guarantee water absorption of roots under salt stress and alleviate the osmotic stress damage.
L51. What is this important role?
Reply:Na+ is a major toxic ion which has the similar ionic radius and hydration energy with K+, competing with K+ for the enzyme binding site. K+/Na+ unbalance motivate physiological metabolism disorder and oxidative stress, which can cause irreversible damage to plants than osmotic stress. Na+ excessive uptake restrains the absorption of Ca2+ and other mineral elements, causing nutritional deficiency. K+/Na+ homeostasis plays an important role in maintaining the physiological activity of cells.
L54. “Two mainly pathways that regulate Na+ concentrations in cytoplasm of plant cells” is a sentence fragment. Please rewrite
Reply:Thank you for your suggestion! The sentence was corrected in the revised MS as the following. There are two main pathways to regulate Na+ concentrations in cytoplasm of plant cells.
L73. Which proteins could MDA inactivate?
Reply: MDA could inactivate enzymatic proteins, carrier proteins, cytoskeletal proteins, mitochondrial, antioxidant proteins, and so on.
3.Methodology
L104. Why did the authors choose 0.7 % of NaCl?
Reply: In our preliminary study, different concentrations of NaCl were used to screen cultivars for salt tolerance, and 0.7% NaCl was determined as the appropriate concentration for the identification of salt tolerance of asparagus. Therefore, this concentration was still used for this experiment.
L179, L180, L181 and elsewhere. Please write cDNA correctly
Reply: Thank you very much for the suggestion! We have revised them in the revised manuscript.
L187. What is the sequence of the primers used in this study? What was the PCR efficiency?
Reply: We have added the sequence of the primers used in this study in the revised manuscript. In addition, the thermal cycle parameters of PCR were 94°C for 10 min, followed by 40 cycles of 94°C for 15 s and 60°C for 1 min in a 20 μL volume.
Table 1: The following primers were used in this experiment
Gene ID |
primer name |
primer sequence(5'-3') |
LOC109820108 |
LOC109820108-F |
CAATGTCAAGGCCAAGATCC |
|
LOC109820108-R |
CTTCTGGATGTTGTAGTCGG |
LOC109836512 |
LOC109836512-F |
CATCATCAGACCTTGAGCAG |
|
LOC109836512-R |
AGGAGGAGAAATTAGGGTTAGG |
LOC109839605 |
LOC109839605-F |
CTCTCCTCTCATCATCTACAC |
|
LOC109839605-R |
CTCTCCTCTCATCATCTACAC |
LOC109837240 |
LOC109837240-F |
TCACTCACGATGTTTCTCAC |
|
LOC109837240-R |
TCAATGTTTCAGGACTCCCA |
4.Results
The “X” and “Y” legends for the figures are somehow small
Reply: We have made the size of the legends larger in the revised manuscript.
L233-234. The sentence needs to be relocated
Reply: We have relocated the sentence in the revised manuscript. The notes of figure are placed after the title.
The figure legends are consistently too short. Additional details and explanations are needed.
Reply: We have added more details and explanations e. g. CK meaning seedlings were watered by Hoagland solution without NaCl, and SS meaning seedlings were subjected to salt stress of 0.7% NaCl.
Figure 6. I could not identify the control group.
Reply: The plots in figure 6 meaning the ratio of relative gene expression level of salt stress to control treatment in two cultivars.
5.Discussion
The results of this study were very nicely discussed
Reply: Thank you very much for the comments and encouragement!
L352. What is this role?
Reply: The role of accumulating organic compounds is consistent to the introduction L43. More accumulation of organic compounds can keep lower osmotic potential of plants and effectively maintain water absorption of roots under salt stress. It can guarantee appropriate physiological metabolism and growth under salt stress.
6.Conclusions
What are the future directions or additional needed research in the area?
Reply: Identification about salt tolerance traits of different asparagus cultivars and key salt tolerance genes, breeding asparagus cultivars for salt tolerance,which all need future research

Reviewer 4 Report
The manuscript deals with a very relevant topic and is well prepared. However, please check the English style, grammar and spelling.
Moreover, please increase the font size in all figures. The legends etc. are really hard to read in the present state.
Another question: Why did you perform the study in the greenhouse? At least in Europe, asparagus is cultivated in the open field and very little in greenhouses...
Regarding line 199: Did you assess root growth by the root dry weight? I am wondering why you mention significant effects as the root dry weight does not show significant diferences between the two sailinity treatments.
Author Response
1.The manuscript deals with a very relevant topic and is well prepared. However, please check the English style, grammar and spelling.
Reply: The grammar and spelling of the manuscript has been checked and revised by an native English speaker.
2.Moreover, please increase the font size in all figures. The legends etc. are really hard to read in the present state.
Reply: Thank you very much for your suggestion! We have increased the font size of coordinate axis titles and legends.
3.Another question: Why did you perform the study in the greenhouse? At least in Europe, asparagus is cultivated in the open field and very little in greenhouses...
Reply: The objective of this study was to explore the response of asparagus seedlings to salt stress. The experiment was conducted under pot sand culture, and the sand was washed with water to keep same level of salt in each treatment. Duration of the test was about a month. It was difficult to accurately control salt level, temperature, and other parameters in field cultivation. So, we performed the study in the greenhouse.
4.Regarding line 199: Did you assess root growth by the root dry weight? I am wondering why you mention significant effects as the root dry weight does not show significant differences between the two salinity treatments.
Reply: Thank you very much for your question! This is a writing error. The root growth showed a slight decrease in JL1 while a slight increase in Apollo was observed under salt stress. The relative biomass showed significant differences between two cultivars, it showed a significant decrease in JL1. We evaluated salt tolerance by the relative biomass.

Round 2
Reviewer 2 Report
The authors considered the suggestions and the comments provide so now the manuscript is ready to be published
Author Response
Thank you very much for the review and suggestion to this manuscript!